



# The Role of Coarse Aerosol Particles as a Sink of HNO₃ in Wintertime Pollution Events in the Salt Lake Valley

Amy Hrdina[1*], Jennifer G. Murphy[1], Anna Gannet Hallar[2], John C. Lin[2], Alexander Moravek[1**], Ryan Bares[2], Ross C. Petersen[2***], Alessandro Franchin[3,4], Ann M. Middlebrook[3], Lexie Goldberger[5****], Ben H. Lee[4], Munkh Baasandorj[2], Steven S. Brown[3]

[1]Department of Chemistry, University of Toronto, Toronto, ON, M5S 0A6, Canada
[2]Department of Atmospheric Sciences, University of Utah, Salt Lake City, UT, 84112, USA
[3]NOAA Earth System Research Laboratory (ESRL) Chemical Sciences Division, Boulder, CO, 80305, USA
[4]Cooperative Institute for Research in Environmental Sciences (CIRES), University of Colorado, Boulder, CO, 80309, USA
[5]University of Washington, Department of Atmospheric Sciences, Seattle, WA, 98195, USA
[*]Now at Department of Civil and Environmental Engineering, Massachusetts Institute of Technology, Cambridge, MA, USA
[**]Now at Department of Chemistry, York University, Toronto, ON, M3J 1P3, Canada
[***]Now at Department of Physical Geography and Ecosystem Science, Lund University, Sweden
[****]Now at Pacific Northwest National Laboratory

Correspondence to: Jennifer G. Murphy (jen.murphy@utoronto.ca)

**Abstract.** Wintertime ammonium nitrate (NH₄NO₃) pollution events burden urban mountain basins around the globe. In the Salt Lake Valley of Utah in the United States, such pollution events are often driven by the formation of persistent cold air pools (PCAP) that trap emissions near the surface for several consecutive days. As a result, secondary pollutants including fine particulate matter less than 2.5 µm in diameter (PM$_{2.5}$), largely in the form of NH₄NO₃, build up during these events and lead to severe haze. As part of an extensive measurement campaign to understand the chemical processes underlying PM$_{2.5}$ formation, the 2017 Utah Winter Fine Particulate Study, water-soluble trace gases and PM$_{2.5}$ constituents were continuously monitored using the Ambient Ion Monitoring Ion Chromatograph system (AIM-IC) at the University of Utah campus. Gas phase NH₃, HNO₃, HCl and SO₂ along with particulate NH₄⁺, Na⁺, K⁺, Mg²⁺, Ca²⁺, NO₃⁻, Cl⁻, and SO₄²⁻ were measured from January 21 to February 21, 2017. During the two PCAP events captured, the fine particulate matter was dominated by secondary NH₄NO₃. The comparison of total nitrate (HNO₃ + PM$_{2.5}$ NO₃⁻) and total NH$_x$ (NH₃ + PM$_{2.5}$ NH₄⁺) showed NH$_x$ was in excess during both pollution events. However, chemical composition analysis of the snowpack during the first PCAP event revealed that the total concentration of deposited NO₃⁻ was nearly three times greater than that of deposited NH₄⁺. Daily snow composition measurements showed a strong correlation between NO₃⁻ and Ca²⁺ in the snowpack. The presence of non-volatile salts (Na⁺, Ca²⁺, and Mg²⁺), which are frequently associated with coarse mode dust, was also detected in PM$_{2.5}$ by the AIM-IC during the two PCAP events, accounting for roughly 5% of total mass loading. The presence of a significant particle mass and surface area in the coarse mode during the first PCAP event was indicated by size-resolved particle measurements from an Aerodynamic Particle Sizer. Taken together, these observations imply that atmospheric measurements of the gas phase and fine mode particle nitrate may not represent the total burden of nitrate in the atmosphere, implying a potentially significant role for uptake by coarse mode dust. Using the NO₃⁻:NH₄⁺ ratio observed in the snowpack to estimate the proportion of



atmospheric nitrate present in the coarse mode, we estimate that the amount of secondary $NH_4NO_3$ could double in the absence of the coarse mode sink. The underestimation of total nitrate indicates an incomplete account of the total oxidant production during PCAP events. The ability of coarse particles to permanently remove $HNO_3$ and influence $PM_{2.5}$ formation is discussed using information about particle composition and size distribution.

## 1 Introduction

Episodes with high particulate matter (PM) pollution occur frequently in urban air basins across the globe during winter months when a stable boundary layer persists for multiple days, including the mountain valleys in the western US (Baasandorj et al., 2017; Bares et al., 2018; Green et al., 2015; Silcox et al., 2012; Whiteman et al., 2014), the Po Valley in Italy (Bernardoni et

al., 2017; Vecchi et al., 2018) , and the Sichuan (Tian et al., 2019) and Twin-Hu (Gao et al., 2019) Basins in China. These mountain basins, including the Salt Lake Valley (SLV) in Northern Utah, experience strong temperature inversions that develop into persistent cold-air pools (PCAP), which suppress vertical mixing and trap emissions within a shallow boundary layer (Lareau et al., 2013; Whiteman et al., 2014). Under these conditions, mass loadings of fine particles smaller than 2.5 µm ($PM_{2.5}$) often reach values of $60 - 80$ µg m$^{-3}$ in the western US, which are far above the U.S. National Ambient Air Quality

Standard (35 µg m$^{-3}$, 24 h average) (Lareau et al., 2013; Silcox et al., 2012; Whiteman et al., 2014). Mass loadings of $PM_{2.5}$ that exceed this standard have been associated with increased risk of mortality, especially cardiopulmonary or cardiovascular disease mortality (Pope et al., 2003, 2017). However, despite significant societal concerns about the impact on human health from these $PM_{2.5}$ pollution episodes and their common occurrence around the world, the major chemical processes that drive PM formation in these regions are still uncertain.

Several studies conducted in Northern Utah have shown that $PM_{2.5}$ during these PCAP episodes is predominantly composed of ammonium nitrate ($NH_4NO_3$), accounting for roughly $60 - 80\%$ of the total dry particle mass (Baasandorj et al., 2017; Franchin et al., 2018; Hansen et al., 2013; Kelly et al., 2013; Kuprov et al., 2014; Long et al., 2002, 2003; Mangelson et al., 1997). $NH_4NO_3$ formation is thermodynamically favorable under conditions with low temperatures and high relative humidity based on equilibrium partitioning with gas phase ammonia ($NH_3$) and nitric acid ($HNO_3$) (Mozurkewich, 1993; Nowak et al.,

2010; Seinfeld and Pandis, 2006), illustrated in the right-hand portion of Fig. 1.

$NH_3$ emissions are often associated with agricultural activities and waste disposal; however, in recent years automotive emissions and industrial processes have become increasingly important sources (Behera et al., 2013; Bishop et al., 2010, 2016; Livingston et al., 2009; Nowak et al., 2012; Roth et al., 2019; Suarez-Bertoa et al., 2014; Sun et al., 2017). In contrast, $HNO_3$ is formed in the atmosphere and is a major sink of nitrogen oxides, which are emitted primarily through fossil fuel combustion.

The two dominant mechanisms highlighted in Fig. 1 that lead to $HNO_3$ occur either from 1) oxidation of $NO_2$ via reaction with OH (in orange), which is photochemically driven during the day, or 2) through heterogenous uptake of $N_2O_5$ (in black), which typically occurs at night. Gas phase $HNO_3$ and $NH_3$ and particulate $NH_4NO_3$ are removed from the atmosphere through dry





and wet deposition under typical atmospheric conditions. However, under PCAP conditions, the stably stratified boundary layer reduces convective mixing, which in turn allows pollutants to accumulate. In addition, $HNO_3$ can also be lost through heterogenous reaction with dust and sea salt components, such as $CaCO_3$ and $NaCl$ (Beichert and Finlayson-Pitts, 1996; Dasgupta et al., 2007; Fenter et al., 1994; Liu et al., 2008a). When the loss of $HNO_3$ to deposition is suppressed under PCAP

conditions, the loss to reactive uptake to airborne dust and sea salt can potentially become more important.

To formulate effective control strategies that reduce wintertime $PM_{2.5}$ in northern Utah, measurement campaigns have been conducted to improve our understanding of fine particulate formation during these PCAP episodes (Baasandorj et al., 2017; Bares et al., 2018; Hansen et al., 2010; Kelly et al., 2013; Kuprov et al., 2014; Lareau et al., 2013; Malek et al., 2006; Mangelson et al., 1997; Silcox et al., 2012; Whiteman et al., 2014). Several analyses of the PM build-up in cold-air pool events in the SLV

have been published, all based on measurements from one or several ground sites (Baasandorj et al., 2017; Kuprov et al., 2014; Long et al., 2003; Silcox et al., 2012). Long-term $PM_{2.5}$ composition measurements in the SLV are made using filter samples (Kelly et al., 2013; Kuprov et al., 2014; Long et al., 2003; Mangelson et al., 1997; Silcox et al., 2012; Whiteman et al., 2014), with $PM_{2.5}$ speciation done on filter extracts in accordance with the U.S. Environmental Protection Agency (EPA) Speciation and Trends Network (STN) protocol. Kuprov et al (2014) was the first to report the gas and $PM_{2.5}$ composition during pollution

events in northern Utah using Ambient Ion Monitoring system coupled with Ion Chromatographs (AIM-IC); however, this study only measured anionic species, providing $HNO_3$ and particle nitrate ($pNO_3^-$) concentrations at hourly resolution. Ambient $NH_3$ concentrations were monitored using a chemiluminescence-based $NH_3$ analyzer, but particulate ammonium ($pNH_4^+$) had to be inferred based on the assumption that both $pNO_3^-$ and particulate sulfate ($pSO_4^{2-}$) were in the form of their respective ammonium salts. Particle mass fractions of crustal and carbonaceous material in $PM_{2.5}$ were calculated on 24 h average

measurements of Al, Si, elemental and organic carbon in fine particles from the NCore program (US EPA, 2010).

Recent research in the SLV has focused on determining whether ammonium nitrate mass loading is most sensitive to reductions in $NH_3$ or $HNO_3$ to inform $PM_{2.5}$ reduction strategies (Kuprov et al., 2014). Baasandorj et al. (2017) recently proposed that $PM_{2.5}$ forms overnight in the upper layers of the PCAP and mixes down to the surface, enhancing the total $PM_{2.5}$ loading experienced in the SLV. This explanation of stratified particulate nitrate chemistry throughout PCAP events suggests a

complex coupling of a $HNO_3$-limited surface layer with a $NH_3$-limited elevated layer (Baasandorj et al., 2017). Aircraft measurements during the 2017 Utah Winter Fine Particulate Study (UWFPS) provided the first detailed vertically-resolved chemical composition of these PCAP episodes in the SLV. Analysis presented in McDuffie et al. (2019) confirms that the formation of $NH_4NO_3$ during these pollution events is largely limited by $HNO_3$, but with more frequent periods approaching $NH_3$-limited conditions in the upper boundary layer over time (McDuffie et al., 2019). Regimes of both $NH_3$-limited and

$HNO_3$-limited conditions are also consistent with an aerosol thermodynamic model sensitivity study of the same data by Franchin et al. (2018), who found simulations of non-refractory $PM_1$ mass were sensitive to reductions in both total ammonium and total nitrate (Franchin et al., 2018). Box model studies of the odd oxygen budget during UWFPS by Womack et al. (2019) assumed that $NH_4NO_3$ formation during PCAP episodes is $HNO_3$-limited, and found full episodes tend to be $NO_x$-saturated. In conditions where $pNO_3^-$ production is $HNO_3$-limited, it is often assumed that reduction in $NO_x$ would directly lead to



reductions in $HNO_3$. However, Womack et al. (2019) showed the production of $HNO_3$, and therefore, $NH_4NO_3$ formation, is sensitive to changes in VOC concentrations emphasizing the complexity of chemical processes involved in PM formation under these conditions.

The balance of factors influencing $HNO_3$ formation are still being understood. Modelling results from both McDuffie et al. (2019) and Womack et al. (2019) predict that up to ~ 50 % of $HNO_3$ production is from the heterogeneous $N_2O_5$ pathway in the residual layer. This is in line with ground-based measurements of $N_2O_5$ by Baasandorj et al. (2017) highlighting the importance of nighttime pathway (Baasandorj et al., 2017). Less attention has been paid to the possibility that alternate sinks for $HNO_3$ may also significantly limit the amount of nitrate that can contribute to $PM_{2.5}$ formation.

Here, results are presented from ground site observations in Salt Lake City from a moderately elevated site at the edge of the SLV during the UWFPS campaign in January and February 2017. Atmospheric concentrations of inorganic trace gases and water-soluble $PM_{2.5}$ constituents were measured by an AIM-IC, equipped with both cation and anion ion chromatographs. During PCAP episodes, we observed the formation and build-up of $NH_4NO_3$ over several days. Between the hours of 8:00 and 18:00 on several PCAP days, we also observed the presence of elevated levels of non-volatile cations in $PM_{2.5}$. To investigate the role of mineral/coarse mode aerosol in limiting the availability of $HNO_3$ during these events, the final section presents estimates of $HNO_3$ lifetime against reactive uptake onto mineral/coarse mode dust based on Aerodynamic Particle Sizer (APS) data and estimates of the total nitrate budget using the chemical composition of the snowpack measured during UWFPS.

## 2 Experimental

### 2.1 Site Description

The SLV (1,300 km$^2$) is a mountain basin in northern Utah surrounded by steep mountain ranges with the Great Salt Lake sitting to the northwest. The Wasatch Mountains (peak at 3,636 masl) span the entire eastern border of the valley with the Oquirrh Mountains (peak at 3,235 masl) to the west. There is a narrow passage in the Southern Traverse Range (1,878 masl) that creates a small opening into Utah Valley. Salt Lake City sits at the base of the valley (1,288 masl), making a vast elevation difference between the metropolis and the surrounding mountains. Measurements were conducted from the rooftop of the William Brown Building 33 m above ground level at the University of Utah (UU) located 40°45'58.7"N and 111°50'51.6"W in Salt Lake City. The UU site is located 155 m above the valley floor along the northeast sidewall of SLV. The site is impacted by local traffic that enters the university on the north side of campus.

### 2.2 Instrumentation

Hourly averages of $PM_{2.5}$ chemical composition and gas phase precursors were measured using an online continuous AIM-IC instrument (Model 9000D, URG Corp, Chapel Hill, NC). Briefly, gases and particles are sampled from the same air flow by passing through a short inlet with elements for size selection ($PM_{2.5}$ impactor), gas collection (wet parallel plate denuder), and particle capture (saturation chamber) at a flow rate of 3 L min$^{-1}$. Aqueous solutions containing analytes originally in the gas





and particle phases are separately transferred through 22-m inlet lines into 5 mL glass syringes that are subsequently analyzed by IC following hour-long collection periods. More in-depth details about the AIM-IC system, including important adaptations from the standard commercial system, can be found in Markovic et al. (2012). During the UWFPS campaign, the components of the inlet assembly were housed in a small weatherproof box 2 m above the roof of the UU site and the IC systems were

housed in a laboratory below. Standard calibrations of the ICs were performed offline using mixed ion standard solutions before and after the measurement period. Charged species ($Na^+$, $NH_4^+$, $K^+$, $Mg^{2+}$, $Ca^{2+}$, $NH_3$, amines, $Cl^-$, $NO_2^-$, $NO_3^-$, and $SO_4^{2-}$) were measured using Dionex ICS-2000 equipped with concentrator columns (TCC-ULP1 and TAC-ULP1), guard columns (CG17 and AG19) and 4 mm analytical columns (CS17 and AS19). Gradient elution methods, using methanesulfonic acid (MSA) and potassium hydroxide (KOH), were carried out using electrolytically regenerated suppressors to reduce the

influence of the eluent in the conductivity detection. Carbonate salts are never quantified in the anion IC due to the natural presence of carbonate in the distilled deionized water reservoir that sustains the IC systems. Background measurements of the entire AIM-IC system were conducted by overflowing the sampling inlet using high purity zero air to determine the method zero and detection limits of each species. The limits of detection (LOD), based on $3\sigma$ of the background, for trace gases were found to be 150 ppt, 20 ppt, 40 ppt and 10 ppt for $NH_3$, HCl, $HNO_3$ and $SO_2$, respectively. The LODs for $PM_{2.5}$ chemical

constituents were 0.2 µg m$^{-3}$, 0.1 µg m$^{-3}$, 0.1 µg m$^{-3}$, 0.04 µg m$^{-3}$, 0.4 µg m$^{-3}$, for $pNH_4^+$, $pNa^+$, $pK^+$, $pMg^{2+}$, and $pCa^{2+}$ and 0.03 µg m$^{-3}$, 0.03 µg m$^{-3}$, and 0.01 µg m$^{-3}$ for $pCl^-$, $pNO_3^-$ and $pSO_4^{2-}$. Traces gases and $PM_{2.5}$ constituents measured as cations are reported from January 21 to February 21, 2017. Due to technical difficulties, anionic constituents measured from 21 January to 8 February were compromised. Therefore, $PM_{2.5}$ anionic components are only reported from 13 to 21 February and the trace gases measured as anions during the former period may be less reliable.

Continuous measurements of total $PM_{2.5}$ mass concentrations were also captured at the site using a Filter Dynamics Measurement System (FDMS) with a tapered element oscillating microbalance ambient particulate monitor (FDMS TEOM 1400ab, Thermo Fisher Scientific) provided by the Utah Division of Air Quality (UDAQ). Ambient air was sampled at ~1.5 m above the roof through ~7 m long ¼" O.D. PFA line with volumetric flow of 20 L min$^{-1}$. A custom-made inertial PM impactor was connected to the entrance of the sampling inlet to remove coarse PM and water from the sample flow. A critical

orifice was installed downstream of the PM impactor to restrict the sample flow maintaining a pressure below 200 mbar. The air sample was sent through an additional filter to remove particles larger than 2.5 µm. An Aerodynamic Particle Sizer (APS; TSI Inc, Model 3321) was operated from 26 January to 10 February to size and detect particles with an aerodynamic diameter between 0.54 and 19.81 µm based on the time-of-flight of the particle between two (633 nm) He-Ne lasers. This instrument was attached to an aerosol inlet that included 2.54 cm diameter stainless-steel tube connected to a stainless-steel rain cap with

approximately a 10 cm diameter. Aerosol mass was calculated using the Aerosol Instrument Manager (AIM) software for the APS Model 3321. The particle density used for calculating the aerosol mass concentration was 1.0 g cm$^{-3}$ and particles were assumed to be spherical. The APS was attached to this inlet system via a custom-made stainless-steel pickoff. To improve aerosol transmission through the inlet, an external pump was attached to the system. The pump pulled at approximately 7.2 L min$^{-1}$. The calculated transmission shows a 50% cut-off at approximately 7 µm for this inlet system (Skiles et al., 2018).



### 2.3 Snow Sampling and Analysis

The inorganic chemical composition ($Na^+$, $NH_4^+$, $K^+$, $Mg^{2+}$, $Ca^{2+}$, $NH_3$, $Cl^-$, $NO_2^-$, $NO_3^-$, and $SO_4^{2-}$) of the snowpack was monitored throughout the campaign during periods of snow cover. The snowpack composition and depth were measured daily within a 2 m radius around the AIM-IC inlet atop UU using a clean graduated 5 cm diameter beveled snow corer. Samples

were collected in triplicate along with a field blank to account for any contamination during sampling. Collected samples were melted in sealed containers. The resulting snow melt solution was measured for total melt volume and pH (Hach SensION™+ PH1 Portable pH meter). Snow melt samples were then filtered through 0.25 µm PTFE filters (Whatman, VWR Distributors) before analysis utilizing ion chromatography.

### 2.3. Aerosol Thermodynamic Modelling using ISORROPIA

Thermodynamic modelling was used to examine the expected gas and particle partitioning of semi-volatile constituents based on the local meteorological conditions and the observed concentrations of gas and particle constituents. These models carry out bulk calculations to estimate the gas phase and particle phase composition of inorganic species, in which all particles are assumed to have the same chemical composition. To examine how the presence of non-volatile particle components observed during this period may affect nitrate partitioning, ISORROPIA v2.1 (Nenes et. al, 1998; Fountoukis and Nenes, 2007) was

used. Model calculations were run in forward mode, with observed total ammonia ($NH_3$ + $pNH_4^+$), particulate sulfate ($pSO_4^{2-}$), total nitrate ($HNO_3$ + $pNO_3^-$), total chloride ($HCl$ + $pCl^-$), non-volatile particulate species ($pK^+$, $pNa^+$, $pCa^{2+}$ and $pMg^{2+}$), along with ambient relative humidity and temperature as inputs. Model runs were conducted using the metastable state option, which prevents solid formation by forcing all particulate components to remain in an aqueous state.

### 3 Results and Discussion

**3.1 Trace Gas and PM$_{2.5}$ Composition in the Salt Lake Valley**

The AIM-IC measurement period included two PCAP pollution episodes, 27 January to 3 February 2017 (episode 1) and 13 to 17 February 2017 (episode 2), defined by the meteorological conditions that stabilize the boundary layer and by the total observed $PM_{2.5}$ mass, separated by a relatively clean period from 5 to 12 February 2017. The PCAP episodes can be clearly seen in the buildup of $PM_{2.5}$ mass and the inorganic constituents measured by the AIM-IC as shown in Fig. 2. Pollution periods

with total 24h average $PM_{2.5}$ mass > 17.5 µg m$^{-3}$ are deemed PCAP periods while periods with a 24h average $PM_{2.5}$ mass ≤ 2 µg m$^{-3}$ are considered clean periods, as previously defined by Whiteman et al. (2014). The pollution period from 27 January to 3 February 2017 (episode 1) was the most persistent and severe during the UWFPS measurement campaign.

Both pollution events are marked in Fig. 2, while the full suite of inorganic gases and $PM_{2.5}$ components was measured by the AIM-IC only from 8 to 19 February 2017. The increase in total $PM_{2.5}$ mass at the onset of episode 1 on 27 January coincides

with a rise in $pNH_4^+$, seen in Fig. 2a, while there is an initial drop in ambient $NH_3$ concentration before it plateaus at mixing



ratio ~ 1 ppb as the PCAP progresses. The $HNO_3$ and $pNO_3$ data, depicted in Fig. 2b for the second pollution event, confirm the major water-soluble component of $PM_{2.5}$ in the SLV is $NH_4NO_3$, as previously reported (Hansen et al., 2010; Kelly et al., 2013; Kuprov et al., 2014; Long et al., 2003). Compared to the $pCl^-$ and $pSO_4^{2-}$ mass loadings measured during that event, Fig. 2b shows the total mass of $pNO_3^-$ is an order of magnitude larger. Particularly at midday, the majority of the $PM_{2.5}$ mass is

$NH_4NO_3$. It is important to note the organic fraction of $PM_{2.5}$ was not measured by the AIM-IC. Based on TEOM total mass and AIM-IC measured mass fractions, the AIM-IC captured 75 % of the total $PM_{2.5}$ mass.

The distinction between PCAP episodes 1 and 2 is important to note because the stability of the cold-air pool and its persistence influences the intensity of $PM_{2.5}$ buildup in the SLV. Episode 1 was the most severe, persisting for nine days and allowing $PM_{2.5}$ daily averages to reach the highest observed through the measurement period, a maximum hourly average of 60 µg m$^{-3}$.

The episode occurred a few days after a storm deposited roughly 22 cm of snow across the SLV and Wasatch front. Ambient temperatures remained below freezing, between 263 and 273 K, sustaining the snowpack throughout the event. This in turn increased the surface albedo, limiting the strength of convection that can disrupt the stable boundary layer (Whiteman et al, 2014). Winds during this episode were light, averaging 1 m s$^{-1}$, with gusts under 7 m s$^{-1}$ during the most stable and calm cold pool period from 31 January to 2 February.

The second PCAP observed during the campaign was much shorter-lived with moderate cold pool conditions only present from 13 through 17 February. This episode was distinctly different than episode 1 with no snow cover and much warmer conditions, with temperatures ranging from 270 to 284 K. The winter snowpack in the surrounding mountain range showed signs of spring melt, which extended the surface area of exposed ground. Winds were slightly stronger, averaging 1.6 m s$^{-1}$ with stronger gusts averaging around 8 m s$^{-1}$ caused by stronger convection. The hourly $PM_{2.5}$ average at the UU site peaked

at 28 ug m$^{-3}$, and elevated PM levels only persisted for 2 to 3 full days, so the intensity and duration of episode 2 was about half that of episode 1.

The presence of non-volatile cations ($pCa^{2+}$, $pMg^{2+}$, $pK^+$ and $pNa^+$), in Fig. 2c, was also observed during PCAP events. Of all the non-volatile cations observed, only potassium increases with the same pattern as the increase in $pNH_4^+$ and total $PM_{2.5}$ mass measured by the TEOM. Potassium salts are often found in primary aerosols formed from biomass or biofuel burning

(Pósfai et al., 2003; Rissler et al., 2006). Wood burning is still a commonly used source of heat during the winter months in the SLV and could be the source of $pK^+$ observed during the severe PCAP episode 1. Aerosol Mass Spectrometer (AMS) data from Franchin et al. (2018) showed traces of levoglucosan, a known marker for wood combustion, during the pollution episode. However, levoglucosan was typically measured at close to background values. Overall, the data reported from AMS measurements indicated the organic material from wood combustion is a not a dominant mass fraction on the regional scale

(Franchin et al., 2018). Recent radiocarbon analysis from ground sites in SLV indicate that fossil fuels were the dominant sources of carbonaceous aerosol during winter, contributing to 88% (80 to 98%) of the black carbon in aerosols and 58% (48 to 69%) of the organic carbon in aerosols (Mouteva et al., 2017). Similarly, the AIM-IC data also suggests that $pK^+$ salts generally compose a small fraction of the total $PM_{2.5}$ mass.



The alkali and alkaline metal components $pCa^{2+}$, $pMg^{2+}$ and $pNa^+$ are commonly found in larger primary particles often emitted from the lofting of mineral dust and salts. These species are often found in mineral aerosols, which typically range from 0.1 to more than 10 µm, partially contributing to $PM_{2.5}$ (Zender, 2003). The coarse mode fraction of these primary particles, with diameters greater than 2.5 µm, have sufficiently large settling velocities that they have much shorter lifetimes in the atmosphere

compared to $PM_{2.5}$. If the non-volatile cations observed in $PM_{2.5}$ are present in the largest fine particles, then their lifetime in the atmosphere is considerably shorter than $NH_4NO_3$ found in the accumulation mode. This may be the explanation for the more pronounced diel pattern of $pCa^{2+}$, $pMg^{2+}$, and $pNa^+$ during the PCAP periods. It is also possible that because vertical mixing is suppressed during PCAP episodes, the UU site is in contact with air from the mineral dust source region to a measurable degree only during midday. During the clean periods, the concentrations of these ions measured by the AIM-IC

are very close to their respective detection limits. $pNa^+$, $pCa^{2+}$ and $pMg^{2+}$ $PM_{2.5}$ components peak at midday during pollution events, suggesting similar and/or collocated sources. Concentrations of $pMg^{2+}$, which is an effective tracer for mineral dust along with $pCa^{2+}$ (Maxwell-Meier et al., 2004), are present throughout the entire measurement period, averaging 0.02 µg m$^{-3}$ and reaching a maximum of 0.15 µg m$^{-3}$. It is known that sea salt is the most common source of $pNa^+$, so examining the Ca-to-Na ratios can suggest the predominant source. Particularly for the SLV region, the Great Salt Lake can also be a source of

sea salt. In PCAP episode 1, Ca-to-Na ratios were below 1, suggesting the presence of sea salts. In contrast, in episode 2 ratios of Ca-to-Na were above 1, suggesting the sources of these particulate components differ between the two PCAP periods. Due to the heavy snow fall during episode 1, road salt was continuously applied to all major roads and walkways throughout the UU campus, which may act as a local source of Na salts during periods of heavy traffic. The lack of snow cover during the latter episode and higher temperatures may suggest some influence of mineral dust from the surrounding ground surface areas,

where sediments and weathered minerals are often Ca-rich (Chesselet et al., 1972; Kassomenos et al., 2012; Kuo et al., 2005). Recent studies have found the Great Salt Lake is quickly receding, suggesting the exposed the lake bed could act as a source of dust and sea salt (Hahnenberger and Nicoll, 2012; Skiles et al., 2018; Wurtsbaugh et al., 2017). However, the bulk $PM_{2.5}$ chemical composition measured by the AIM-IC is not detailed enough to unambiguously identify the source of non-volatile cations observed during PCAP events.

**3.3 Estimating HNO₃ Loss to Coarse Mode Particles**

Gas phase $HNO_3$ exists in equilibrium with semi-volatile nitrate in $PM_{2.5}$ and the extent of gas-to-particle partitioning depends on temperature, relative humidity, and $NH_3$, via the formation of $NH_4NO_{3(s)}$ in dry particles or its influence on particle acidity in deliquesced particles. At the same time, $HNO_3$ may undergo net uptake by primary particles when it reacts with their constituents, for example $CaCO_3$ or $NaCl$, to form non-volatile nitrate salts. The rate of $HNO_3$ loss can be impacted by the

composition of the reactive salt and the reactive surface area available (Seinfeld and Pandis, 2006). During the two PCAP episodes, we observed elevated midday concentrations of $pNa^+$, $pCa^{2+}$ and $pMg^{2+}$ in $PM_{2.5}$ using the AIM-IC.



### 3.3.1 Inferences from Aerodynamic Particle Sizing

Coarse mode particles (diameters > 2.5 µm) were not speciated during the UWFPS campaign. There are very few reports of coarse mode nitrate in winter conditions that could offer an approximation of how much $HNO_3$ may have been taken up by coarse mode particles in the SLV. Hansen et al. (2010) is the only study commenting on coarse mode nitrate in northern Utah region, in Lindon, ~ 50 km located south of SLV, during January to February 2007. They observed high mass loadings, up to 80 µg m$^{-3}$ of coarse aerosol ($PM_{10} - PM_{2.5}$), during PCAP events, based on measurements from two GRIMM optical particle counters (Hansen et al., 2010). The authors did not report any quantification of the chemical components of the coarse aerosol collected by integrated filter samples, although they commented that the nitrate levels were relatively low.

To obtain information about the coarse mode loadings during PCAP events in UWFPS, APS data collected during the pollution episode from 28 January to 3 February is displayed in Fig. 3. The particle size bins are colored by mass normalized to the size bin (dM/dlogDp), clearly showing elevated midday burdens of coarse mode particles, which become more pronounced during the intensive cold-air pool period from 30 January to 2 February 2017. This is consistent with the midday increases in coarse mode mass observed across SLV, shown in Fig. 4. The coarse fraction of $PM_{10}$ measured at Hawthorne Elementary (the air quality regulatory site at the base of SLV operated by the UDAQ) shows a similar trend to the APS mass loading of $PM_{10}$-$PM_{2.5}$ observed at UU. In contrast, the Rose Park site (another UDAQ site), which is close to Great Salt Lake, exhibits moderate increases in coarse mode mass compared to UU and Hawthorne. This difference is more evident during the calmest period of the pollution event, when the lowest wind speeds were observed, which suggests the sources of coarse particles are highly localized during PCAP events. An analysis of coarse particles by Li et al. found only moderate correlation between $PM_{10-2.5}$ mass loadings and wind speed at several sites across the U.S (Li et al., 2013).

The time series of the total surface area of PM larger than 2.5 µm in µm$^2$ cm$^{-3}$ (Fig. 3) shows the available coarse mode surface area exceeds 1500 µm$^2$ cm$^{-3}$ for several hours on each of the three most severe pollution days, 30 January to 1 February 2017. For comparison, the total surface area of the fine mode particles is also shown, which is roughly three times the maximum surface area of the coarse fraction, but with less diurnal variability. However, the fine mode aerosol nitrate is assumed to be in equilibrium with gas phase $HNO_3$, in contrast to the coarse mode surface area, which could be viewed as a reactive sink allowing net uptake. The midday increases in coarse particle surface area, therefore, could represent an important permanent sink for $HNO_3$ if the particles are composed of reactive salts such as $CaCO_3$ or NaCl.

A lower limit to the lifetime of $HNO_3$ with respect to uptake by the coarse aerosol surface can be calculated to infer the potential importance of mineral/coarse aerosol particles during these heavy pollution events. The approximate first-order removal rate of $HNO_3$ due to the heterogeneous reaction with coarse aerosol, $k_{het}$, depends on its average molecular speed, $c$, the surface area density of coarse aerosol, $S_A$, and the uptake coefficient, $\gamma$, given in the following expression (Ammann et al., 2013; Crowley et al., 2010; Kolb et al., 2010; Tang et al., 2017):

$$k_{het} = 0.25 \cdot c \cdot S_A \cdot \gamma$$


The variation in minerology of coarse mode particles can influence the efficiency in $HNO_3$ reactive uptake via the uptake coefficient, $\gamma$. This is represented in the range of reported uptake coefficients of $HNO_3$ to $CaCO_3$ ($\gamma = 0.07$ at RH = 50%) and NaCl ($\gamma = 0.11$ at RH = 55 %) (Liu et al, 2008; Saul et al, 2006). The average RH at peak coarse mode loading was 54%; therefore, the reported uptake coefficients close to this RH value were used in our calculations. Literature reports have shown

RH significantly enhances the reaction probability of $HNO_3$ uptake with mineral dust and sea salts (Liu et al., 2008a; Saul et al., 2006; Vlasenko et al., 2006). The detailed kinetics of $HNO_3$ reacting with $CaCO_3$ particles in a flow reactor by Liu et al. (2008) found low uptake values of $\gamma \leq 0.003$ at 10 % RH while at 80 % RH, $\gamma$ can be $\geq 0.21$. This agrees with reports from Vlasenko et al. (2006), in which $HNO_3$ uptake on dust aerosol containing ~5 % $CaCO_3$ was also enhanced by an increase in RH. Saul and coworkers (2006) also found RH affects the uptake of $HNO_3$ onto pure NaCl showing a maximum $\gamma = 0.12$ at

50 % RH while at 85 % RH $\gamma = 0.05$. They also observed an increase in reactivity with the presence of $MgCl_2$ in NaCl powder. The greater reactivity of $MgCl_2$ is consistent with the hygroscopic character of magnesium salts that facilitate the increase in adsorbed water at the particle surface, which can in turn increase the effective uptake of $HNO_3$.

Because $pNa^+$ and $pCa^{2+}$ were the dominant non-volatile species that showed common behavior with the coarse mode mass loading, the rates of uptake with respect to NaCl and $CaCO_3$ were calculated to approximate how quickly $HNO_3$ might be

sequestered. Literature values for the uptake coefficients of $HNO_3$ on NaCl (0.11 at 55 % RH) and $CaCO_3$ (0.07 at 50 % RH) were used to estimate a lifetime against reactive uptake based on the measured coarse mode surface area (Fenter et al., 1994; Liu et al., 2008a; Vlasenko et al., 2006). The timeseries of $k_{het}$ from 27 January to 4 February 2017 is presented in Fig. 5. The effective lifetime of $HNO_3$ against reactive uptake to $CaCO_3$ during episode 1 ranges from 2 to 90 minutes with the shortest lifetimes during periods of peak coarse mode loading. The lifetime against NaCl (0.11) ranges from 1.3 to 57 minutes. These

lifetime calculations assume that the entire surface is reactive and do not account for any mass transfer limitations that may occur on the particle surface. Therefore, the time range is an estimate of the upper limit to the rate of $HNO_3$ uptake, or how quickly the reaction could potentially occur.

Under high RH conditions, which often occur during PCAP events, it is also possible coarse particles may be deliquesced, allowing exchange between surface and bulk ions within a particle. This would result in an increased loss of nitrate if uptake

is not limited to the surface. If the total mass of coarse particles measured during episode 1 is assumed to be composed of $Ca(NO_3)_2$, in which carbonate has been completely displaced, that would amount to an average of 6.2 µg m$^{-3}$ of $HNO_3$ sequestered with a maximum of 32 µg m$^{-3}$ during midday. This assumes $HNO_3$ can react with the bulk particle components, thereby representing an upper estimate of $HNO_3$ mass loss to $PM_{10}$. However, due to the size cutoff of the APS, with significant transmission losses above 10 µm, it is possible that the mass of coarse mode particles is underestimated under some conditions.

The calculations of the removal rate do not include changes in reactive uptake associated with the changes in hygroscopicity of the resulting nitrate salts formed. The presence of calcium or magnesium nitrate salts can enhance the absorbance of water, which has been shown to increase the relative uptake of $HNO_3$ because it is no longer limited to the particle surface, so the estimation of $HNO_3$ loss can also be highly variable (Beichert and Finlayson-Pitts, 1996; Goodman et al., 2000). This is dependent on the deliquescence relative humidity (DRH) of the nitrate salt formed, in which DRH of $Ca(NO_3)_2$ is 10 % at 298



K (Liu et al., 2008b; Sullivan et al., 2009a, 2009b) and NaNO$_3$ is 81 % at 273 K (Seinfeld and Pandis, 2006). The DRH is also temperature-dependent and tends to increase with decreasing temperature. The temperatures during episode 1 were consistently below 273 K, therefore, it is assumed DRH would be higher. The inference from the heterogeneous uptake calculation suggests that the loss of HNO$_3$ can be very rapid. The question of how this loss in HNO$_3$ to coarse aerosol could be shifting the gas-to-

particle equilibrium that exists between HNO$_3$, NH$_3$ and NH$_4$NO$_3$ is still unknown. Therefore, further investigation of time-resolved coarse aerosol chemical composition, in addition to PM$_{2.5}$ composition, is needed to better quantify how much coarse aerosol may be limiting HNO$_3$ availability during these PCAP pollution events and elucidate its effect on NH$_4$NO$_3$ formation. Studies from coastal sites provide evidence that the uptake of nitrate to coarse model aerosol can be substantial. For example, single particle measurements of sea salt particles using ATOFMS in California found that the chloride mole fraction could

decrease from 0.3 to 0, while the nitrate mole fraction could increase from 0 to 0.5 following exposure of sea salt to urban air pollution (Gard et al., 1998). In coastal Florida, measurements using an online ion chromatographic technique showed that deliquesced sea salt particles could have more than 50% of the chloride content replaced by nitrate (Dasgupta et al., 2007). Measurements from a size-resolved integrated sampler showed that nitrate peaked in the coarse mode, where its concentration was close to that of chloride (on a molar basis), and half that of sodium. Size-resolved particle composition measurements near

the coast in British Columbia during the summer showed that during the day, nitrate was only present in coarse mode particles (corresponding to a deficiency in chloride) whereas during the night nitrate could also be measured in accumulation mode particles (Anlauf et al., 2006).

### 3.3.2 Evidence from Snow Analysis

During the campaign period, a snow event occurred prior to the first PCAP episode (January 21), in which the snow remained on the surface until February 3. This provided a snowpack that was subjected to the first pollution event, providing a record of chemical composition changes caused by PCAP exposure. A snow event that happened later in the season occurred during clean conditions (23 February to 2 March 2017), when no PCAP was observed, thus providing a representative example of chemical composition changes in snow not associated with PCAP exposure.

Chemical analysis of the snowpack exposed to PCAP episode 1 reflects a combination of cloud water composition and the atmospheric gas and particle composition of species scavenged by snow fall or later deposited to the snow, including particles larger than the PM$_{2.5}$ cutoff monitored by the AIM-IC. Figure 6 shows a scatter plot of the NH$_4^+$ versus NO$_3^-$ measured in the snowpack, where each data point is representative of the averaged chemical composition of the entire snowpack (sampling area 0.008 m$^2$) on a single day, and in the atmosphere by the AIM-IC (gas phase + PM$_{2.5}$), where PM$_{2.5}$ NO$_3^-$ is assumed to be

equivalent to 90% of PM$_{2.5}$ NH$_4^+$ on a molar basis, and Twin Otter aircraft (gas phase + PM$_1$). Aerosol particle measurements from the aircraft do not include non-volatile cations and their associated nitrates which the AMS does not measure (Franchin et al., 2018). Gas phase HNO$_3$ measurements collected on the aircraft are from the University of Washington HRToF-CIMS instrument operated similarly as described in Lee et al. (2018). Gas phase NH$_3$ measurements from the Twin Otter were taken



using a QC-TILDAS instrument (Aerodyne Inc, MA, USA) and are described in Moravek et al., 2019. Based on aircraft data (Franchin et al., 2018) and AIM-IC measurements at the ground site, the sum of nitrate in the gas phase and particle phases, or total $NO_3$ ($TNO_3$) per $m^3$ of ambient air, is less than the sum of ammonia in the gas phase and particle phases, total $NH_x$ per $m^3$ of ambient air. Therefore, we might expect that there would be more ammonium depositing into the snow compared to

nitrate. However, in Fig. 6, it is clear that the deposition of $NH_x$ is less than half that of total $NO_3$ on a molar basis for the highest deposition amounts. The average $TNO_3:NH_x$ molar ratio of 3:1 measured in the snowpack is significantly different compared to what is observed in the atmospheric measurements. This implies that there could be another source of nitrate being deposited to the snow that is not reflected in the atmospheric measurements of the gas phase and fine mode particles. Further investigation into the composition of the snowpack reveals there is a strong correlation between the amount of $NO_3^-$

and the amount of $Ca^{2+}$ ($r^2 = 0.836$), seen in Fig. 7. Both $Na^+$ and $Ca^{2+}$ are also found in the snowpack in higher abundances than $NH_4^+$, despite being present at much lower levels in $PM_{2.5}$. The larger concentrations of $pNa^+$ relative to $pCa^{2+}$ seen in the AIM-IC $PM_{2.5}$ data are consistent with the larger concentrations of $Na^+$ measured in the snowpack relative to $Ca^{2+}$ concentrations.

This suggests the coarse mode aerosols during this PCAP event were Na-rich. Due to the large excess of µmols of $Na^+$ when

compared to $NO_3^-$, the amount of $Cl^-$ was examined to determine whether Na-rich particles came in the form of NaCl salt. The snowpack showed the highest concentrations of $Cl^-$ out all of the inorganic anions measured, over ten times more than $NO_3^-$ and $SO_4^{2-}$ combined. The total water-soluble ion balance measured in the snowpack (Fig. S1a) shows a slight excess of anions, partially due to the high $Cl^-$ content. However, the concentrations of $Cl^-$ measured in the snow (Fig. S1b) were greater than what can be accounted for by observed $Na^+$. The low concentrations of $Mg^{2+}$ in the snowpack had no noticeable correlation

with $Cl^-$. A closer look at the daily inorganic composition changes and decrease in snowpack height, shown in Fig. S2a, reveal $Ca^{2+}$ and $NO_3^-$ exhibit similar trends throughout the lifetime of the snowpack. This is also true for changes in $K^+$ and $Cl^-$ total snow column concentrations (Fig. S2b).

The proportions of non-volatile cations, where $K^+>Na^+>Ca^{2+}>Mg^{2+}$, found in the snow exhibit a similar distribution observed in $PM_{2.5}$ composition. We infer that our measurements of these constituents in $PM_{2.5}$ also reflects a larger, unmeasured,

contribution of these non-volatile cations in the coarse mode. In addition, the nitrate present in the snow appears to have a strong correlation with $Ca^{2+}$, suggesting there could be a significant amount of particle nitrate in the coarse mode that is not accounted for in $PM_{2.5}$ measurements. This underestimation of particle nitrate during peak coarse mode periods that occur during the day could lead to an underestimation in the production rate of nitrate due to photochemistry. This could impact estimates of the sensitivity of $PM_{2.5}$ to $NH_3$ and $NO_x$ emissions reductions.


### 3.3.3 ISORROPIA Analysis

The comparison of ISORROPIA model runs of predicted $NH_4NO_3$ (µg $m^{-3}$) with and without non-volatile components measured by the AIM-IC are depicted in Fig. 8 through pollution episode 2 from 13 to 19 February 2017 when both anion and





cation gas and particle data was available. ISORROPIA calculates the equilibrium concentrations of the $NH_4^+$-$SO_4^{2-}$-$NO_3^-$-$Na^+$-$Cl^-$-$K^+$-$Mg^{2+}$-$Ca^{2+}$-$H_2O$ system. Overall, the difference in model outputs for these two input conditions demonstrates the influence that the presence of non-volatiles ($pNa^+$, $pCa^{2+}$, $pMg^{2+}$ and $pK^+$) has on the thermodynamic equilibrium between $HNO_3$ and its particulate counterpart $pNO_3^-$.

Previous studies have shown the inclusion of non-volatile cations in thermodynamic models can more accurately reproduce gas-particle partitioning observations by correctly reflecting the ion balance and the ammonium-sulfate ratio (Guo et al., 2018). With the addition of $pCl^-$, $pNa^+$, $pCa^{2+}$, $pMg^{2+}$ and $pK^+$ in ISORROPIA model runs, the model predictions consistently underestimate the $NH_4NO_3$ concentrations observed. This result is expected as the implicit assumption in the ISORROPIA run is that all the non-volatile cations are associated with particle nitrate, which is unlikely to be the case. The model outputs imply

that some non-zero fraction of the nitrate in the gas+$PM_{2.5}$ system is associated with non-volatile cations.

Figure 8 also demonstrates how much more $NH_4NO_3$ would be formed in the fine mode if $HNO_3$ was not sequestered by the coarse particles. To generate the output depicted in blue squares, the total nitrate input into ISORROPIA was calculated based on the $NO_3^-$ to $NH_4^+$ ratio observed in the snowpack during the first PCAP event. As seen in Fig. 6, the $TNO_3^-$ concentration was on average 2.5 times greater than the $TNH_4^+$ concentration measured in the snowpack. This ratio is strictly for the

concentrations measured during the PCAP event. When averaging $TNO_3^-$:$TNH_4^+$ molar ratios for the full lifetime of the snowpack, the average ratio is slightly greater (3). This average molar ratio of 2.5 measured during the pollution event was applied to the total $NH_x$ measured by the AIM-IC and used as the total nitrate input into ISORROPIA along with $pCl^-$, $pNa^+$, $pCa^{2+}$, $pMg^{2+}$ and $pK^+$ concentrations observed by the AIM-IC. Overall, the model predicts greater amounts of $NH_4NO_3$, and therefore, $PM_{2.5}$ to form. When observed $NH_4NO_3$ concentrations are above 10 µg m$^{-3}$, the model predicts roughly two times

more $NH_4NO_3$ mass to form when nitrate lost to coarse particles is accounted for. The additional modelled $pNO_3$ mass is on average 6.8 µg m$^{-3}$ with a maximum of 18 µg m$^{-3}$, which is within the upper limit range of total nitrate loss predicted when assuming uptake to the entire bulk of the aerosol particle. This suggests the coarse particles have the potential to control $NH_4NO_3$ formation indirectly by acting as a permanent sink for $HNO_3$, thereby reducing the amount of available $HNO_3$ to form $PM_{2.5}$.

## 25    4 Conclusion

The AIM-IC measurements confirmed the dominance of $NH_4NO_3$ in wintertime $PM_{2.5}$ pollution events in SLV. The detection of non-volatile cations ($pNa^+$, $pK^+$, $pCa^{2+}$, and $pMg^{2+}$) in $PM_{2.5}$ by the AIM-IC suggests the presence of mineral dust and/or salt during PCAP pollution events, which can potentially impact the availability of $HNO_3$. This is due to the reactive uptake of $HNO_3$ onto NaCl and $CaCO_3$-containing particles, which can be predominantly found in the coarse mode. This can introduce

a permanent sink for $HNO_3$. Due to the lack of coarse mode speciation, auxiliary data was used to estimate the amount of $HNO_3$ potentially lost to reactive uptake. The APS surface area data was used to calculate a lifetime for $HNO_3$ with respect to uptake that was on the order of minutes, depending on the uptake coefficient used, which is influenced by particle composition



(Fenter et al., 1994; Liu et al., 2008; Vlasenko et al., 2006). The potential presence of coarse mode nitrate was also supported by the strong correlation between total $Ca^{2+}$ and total $NO_3^-$ observed in the snowpack. Total $NO_3^-$ in the snowpack was 2.8 times larger than the total $NH_4^+$ implying there is additional $NO_3^-$ that is not observed by $PM_{2.5}$ $pNO_3^-$ and gas phase $HNO_3$ measurements alone.

The equilibrium dynamics were explored by thermodynamic modelling of trace gas and $PM_{2.5}$ composition, during episode 2. Despite the discrepancies between observed and modelled $NH_4NO_3$ concentrations during PCAP episode 2, the inclusion of coarse nitrate, from snowpack estimates, into the model shows that coarse particles could be limiting a significant portion of $HNO_3$ being generated, preventing its reaction with $NH_3$ to form $NH_4NO_3$. The extent to which coarse mode aerosol is limiting $PM_{2.5}$ formation remains unanswered and warrants further investigation of coarse mode composition and the role these particles

play in the atmospheric chemistry during pollution events.

*Data availability*. Groundside and aircraft data from UWFPS can be found here: https://www.esrl.noaa.gov/csd/groups/csd7/measurements/2017uwfps/data.html.

*Competing interests*. The authors declare that they have no conflict of interest.

**Acknowledgements**

The authors like to thank all the members of the UWFPS campaign. The authors would also like to thank many members from the University of Utah in the Atmospheric Sciences department and for their support. We would especially like to thank Catherine Chachere and Lauren Zuromski for processing the APS data.   The authors acknowledge the funding support from

the Michael Smith Foreign Study Supplements Program awarded by the Natural Sciences and Engineering Research Council of Canada. NOAA acknowledges support for Twin Otter flights from the Utah Division of Air Quality under agreement number 16-049696.

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





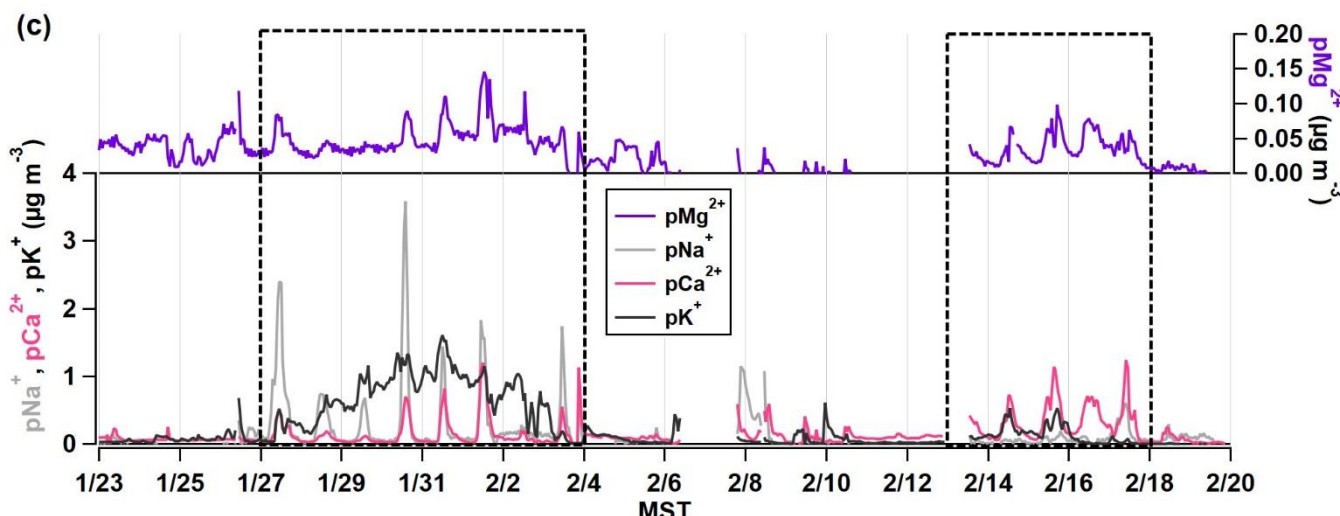

**Fig. 2. (a) The timeseries of NH₃ and pNH₄⁺ observed by the AIM-IC and total PM₂.₅ mass loading (µg m⁻³) measured by the TEOM from 23 January to 19 February 2017. (b) The AIM-IC observations of inorganic gases HCl, HNO₃, and SO₂ and PM₂.₅ components pCl⁻, pNO₂⁻, pNO₃⁻, pSO₄²⁻ present in PM₂.₅. (c) The AIM-IC observations of the non-volatile inorganic cations pNa⁺, pK⁺, pMg²⁺ and pCa²⁺ measured in PM₂.₅ for the entire campaign period. The two PCAP pollution episodes captured during the campaign are emphasized by the dotted black line. Time is expressed in mountain standard time (MST).**



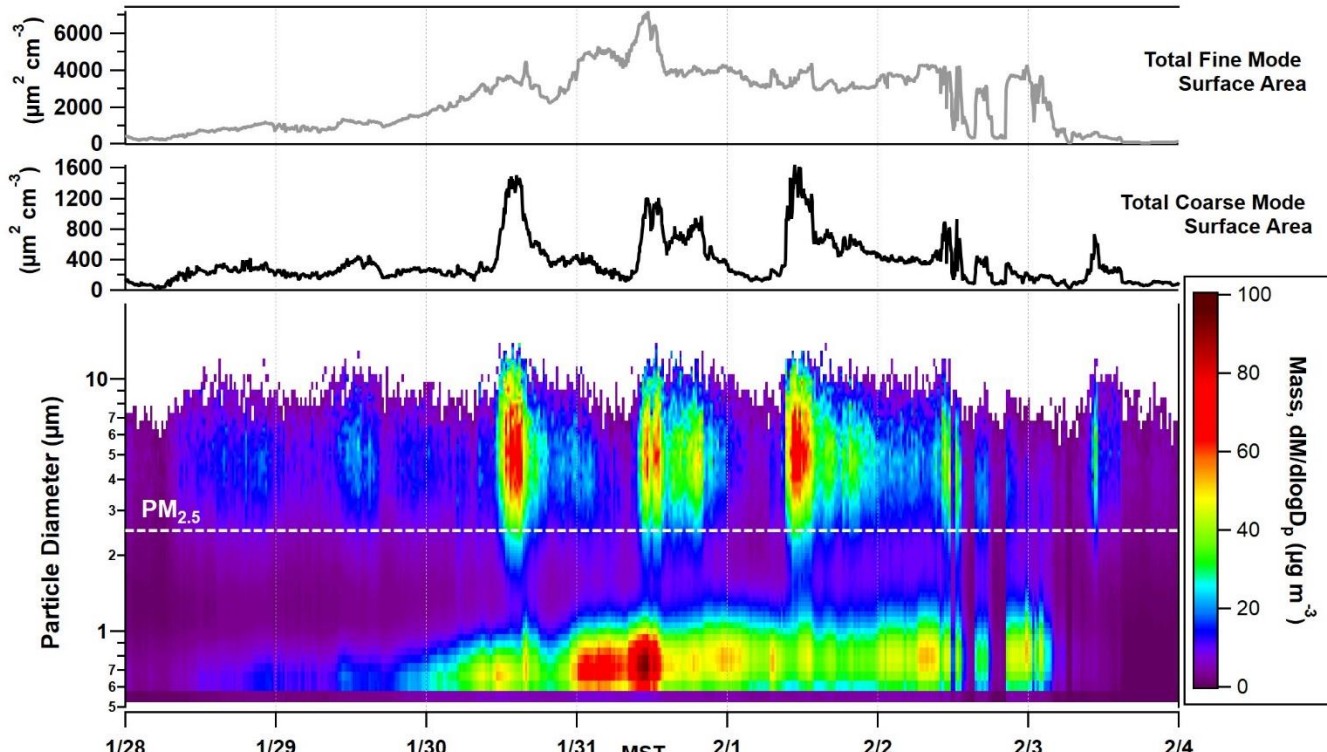

**Fig. 3. (Top) The total fine mode (PM₂.₅) and coarse mode (> PM₂.₅) surface area calculated in µm² cm⁻³ from APS data. (Bottom) The particle size distribution as a function of time, January 28 to February 4, with particle size displayed along the y-axis and colour contours representing mass that is normalized by size bin (dM/dlogDp). The population of particles that the AIM-IC captures is below the labelled PM₂.₅ line.**



**Fig. 4. (a) Wind speed and temperature (° C) measured at UU. The coarse particle (PM$_{10}$ – PM$_{2.5}$) mass loading in µg m$^{-3}$ as a function of time, January 28 to February 4 2017, at three separate ground-based sites: Rose Parks (b) shown in green and Hawthorne Elementary (c) shown in black, which are EPA monitoring sites, and the UU site (d) in blue.**





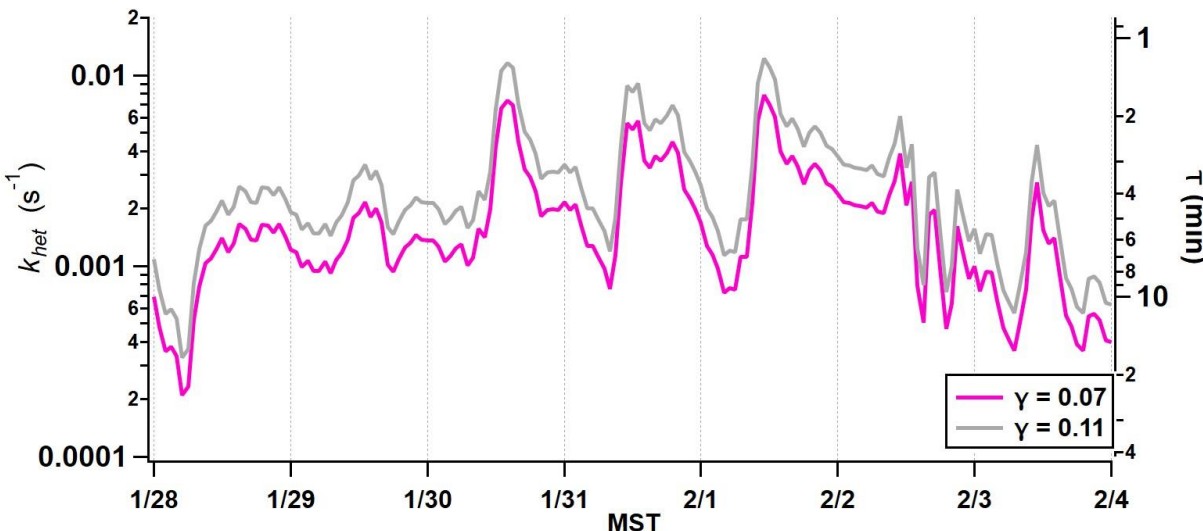

**Fig. 5. Predicted HNO₃ loss rate to coarse particle surface area concentration measured by the APS from 28 January to 4 February 2017.**

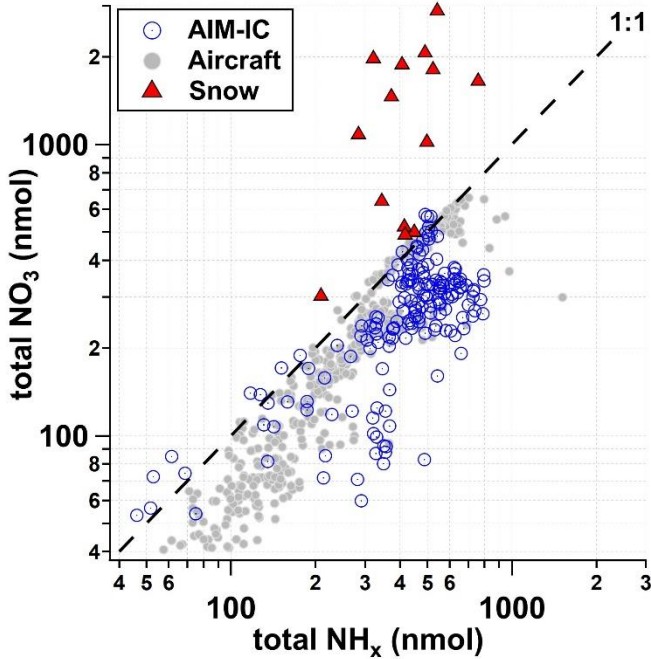

5   **Fig. 6. Total NO₃ (pNO₃⁻ + HNO₃) with respect to total NHₓ measured and predicted from AIM-IC data (grey circles), aircraft measurements (open blue circles) per m³ of ambient air and snow (red triangles) within 0.008 m² sampling area from 21 January to 3 February 2017.**


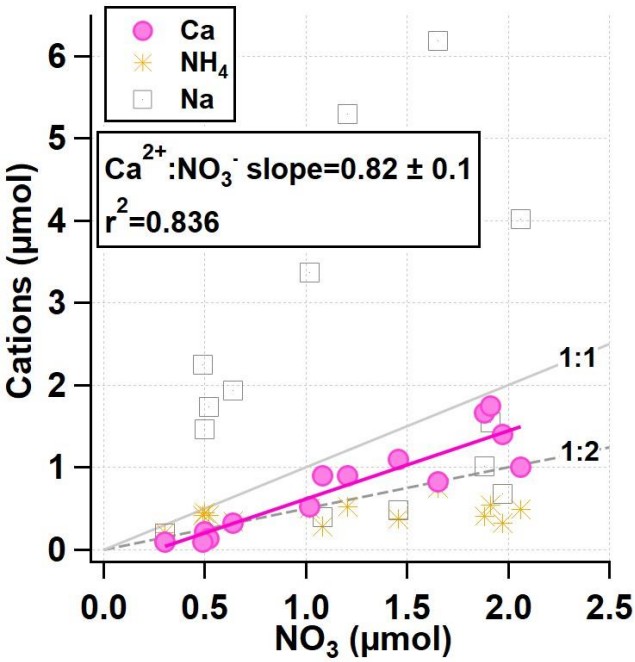

Fig. 7. Daily total μmols of NH₄, Ca and Na vs NO₃ within the snow from 21 January to 3 February 2017, leading into the most severe PCAP period 27 January to 4 February 2017.

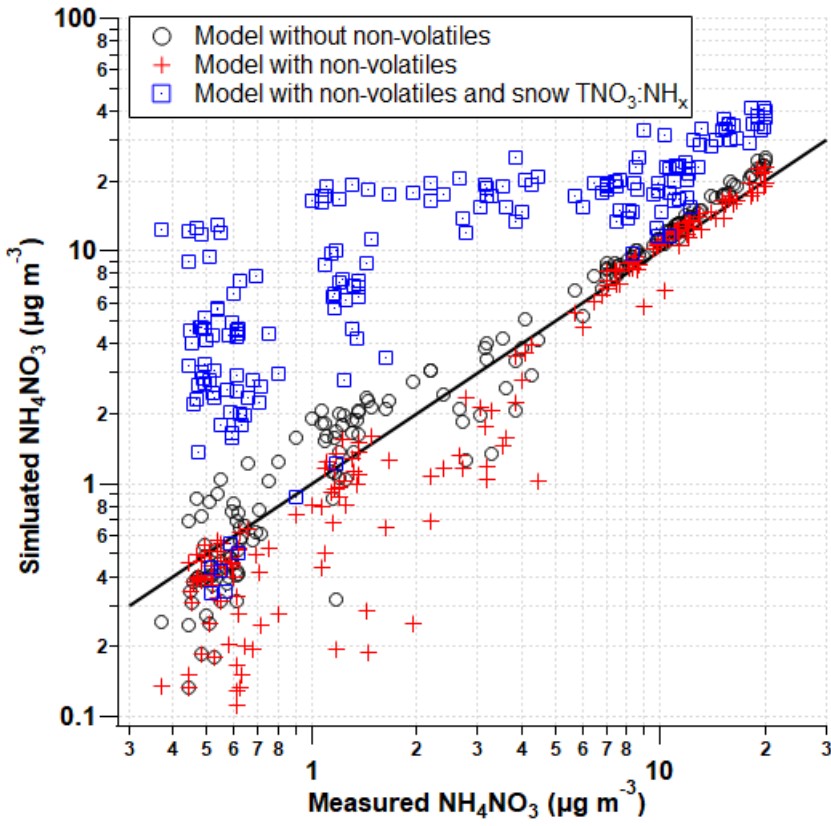

**Fig. 8. Predicted (ISORROPIA) with respected to measured (AIM-IC) ambient concentrations of NH₄NO₃ in µg m⁻³ through the second pollution event from 13 to 19 February 2017. The model was run in three separate conditions; (1) with (red cross) and (2) without (open black circle) the addition of pK⁺, pCa²⁺ and pMg²⁺, and (3) with non-volatiles and the increase in total nitrate based on the ratio of NH₄⁺ to NO₃⁻ measured in the snow.**