# Peer review of "The Role of Coarse Aerosol Particles as a Sink of HNO3 in Wintertime Pollution Events in the Salt Lake Valley"

_Atmospheric Chemistry and Physics, 2020_

## Referee Comment (RC1) · Anonymous Referee #1 · 30 Mar 2020

This paper analyzes data from pollution events in Salt Lake City during cold periods when strong inversions lead to generation of PM2.5 ammonium nitrate. The analysis is limited in that there was no overlap in all instruments during periods of interest. For example, the APS measuring coarse mode size distributions was only operational during a short period of the overall study. It did coincide with a PM2.5 event, but during that event there was no PM2.5 aerosol data. This makes much of the analysis more speculative. Overall, the analysis is somewhat obtuse, especially the thermodynamic analysis, which could use more explanation and additional details. It would benefit from a more rigorous approach in which particle pH and partitioning of ammonium nitrate is discussed. There are some other clarifications needed, which are discussed below.

[Figure]

Specific Comments.

Is the measurement site representative of the greater SLC area? Does its elevation influence this?

Pg 2, Lines 15 to 16 regarding the health effects of NH4NO3. The logic is NH4NO3 drives PM2.5 mass, PM2.5 mass is associated with adverse health and is regulated, so NH4NO3 has adverse health implications and should be reduced to meet PM2.5 regulations. This is standard logic and so a reasonable statement, but if the authors really want to be precise, there is debate if NH4NO3 is toxic. The authors could easily add more details since health studies have bee performed for the regions they are measuring, such as [Watterson et al., 2007]. The results are not as clear as the statement in this paper. Also, why not discuss an environmental effect that can be directly linked to NH4NO3, the effect on visibility and haze due to its hygroscopicity, and what about nitrogen deposition; the latter seems especially important since this paper is really about deposition?

Pg 5 line 3, could not find the Markovic ref. Is the chemistry of the water for the wet denuder and aerosol collector altered to help adsorb species? This is most likely an issue for the wet denuder that is collecting both anions and cations gases. That is, is an absorbing species added to liquid or is it pure water?

What is the RH of the aerosol sampled by the TEOM and APS? Are the particles dry?

Fig 1 is a nice schematic.

It might be more insightful to plot the gases in units of ug/m3 to allow direct comparison with aerosol concentrations, instead of mixing ratios. Although mixing ratios are more traditional.

It should be noted in Fig 3 caption or the text that the total fine mode surface area spans sizes 0.54 to 2.5 um since it was measured by the APS. Is that really total fine particle surface area? Are the particles dry?

Pg 9, the equation for uptake of HNO3 by coarse mode particles seems incorrect. It looks to be the formula for free molecular regime uptake, which does not apply to coarse mode particles. A correction factor due to diffusional resistance, ie something that can be derived from the size distribution and some transition regime mass transfer formula, like Fuchs Sutugin, see Seinfeld and Pandis, may be needed.

Pg 13, line 8-9, This result is expected since.... Please clarify. It might be better to compare the predicted gas and particle partitioning, not NH4NO3 predicted, which is the form that is being assumed that the model is predicting (ie, the model does not output the NH4NO3 concentration it outputs NH4+ and NO3-, along with other species, so I assume the authors are simply taking the output NO3- adding the NH4+ and comparing to a similar addition of data in the form of NH4+ and NO3- ). Aerosol pH seems to largely ignored in this work; pH values are never give despite it controlling the concentration of NO3-. It would be curious, for example, to know what the difference in pH is for with and without cation inclusion. This, and a focus on NO3- predicted vs observed would help interpret the results and provide more detail than the statement in the paper (line 8-9). It seems that what is being implied here is the model is simply doing an ion balance with the input anion and cations; ie if you add more cations than there will be less predicted NH4NO3 since there is some Ca2(NO3)2 (as an example). But isn't it much more complex since adding cations changes the pH which affects the partitioning of NH4+ and NO3-, (and other semivolatile ionorganic species, such as Cl-....). At the higher concentrations the simulated and measured agree. Why is this happening? Is it consistent with the interpretation of what is happening at lower concentrations? The authors might also consider the assumption of mixing state, they implicitly assuming that everything is internally mixed.

The last part of the main text is also not clear (Pg 13, lines 11 to end of section). As I understand it, the idea here is that if all the NO3- and NH4+ found in snow (which got there by reacting with large particles that then fall out) was instead in the gas phase or in fine particles, and PM2.5 nonvolatile cations were present, (but no coarse mode),

then the predicted NH4NO3 would be much higher (2x). But it is not clear what this proves. The situation is much more complex, and really requires a full model. Eg, would not the deposition by gases and fine mode particles change between these two cases. This is not considered. The deposition of the gas species of NH3 and HNO3 is x10 higher than a fine particle, in the case of no coarse mode, the gas concentrations could by much higher and so gas deposition much higher, which could limit PM2.5 NH4+ and NO3- formation. This is not considered in the analysis presented. The general conclusion that the coarse mode is a sink for Total NO3 is clear, but how that impacts PM2.5 NH4NO3 concentrations (it may be better to just talk about NO3- concentrations), is not clear from this simplistic analyses.

---

## Referee Comment (RC2) · Anonymous Referee #2 · 12 May 2020

Hrdina et al. present an interesting, although circumstantial, case for the importance of coarse particles as a sink of HNO3 in the wintertime in the Salt Lake Valley (SLV). The authors do not have direct measurements of coarse particle composition, but make use of PM2.5 composition measurements, coarse particle size distributions, and snowpack chemistry to make a case for reactive uptake of HNO3 on coarse particles influencing the budget of HNO3 in the SLV and associated implications for fine particle chemistry. The manuscript is overall well written and most of the analyses presented sound.

Some of the authors' planned measurements have missing data (e.g., the PM2.5 anions during the main PM episode studied). This, along with the absence of any coarse

particle PM composition measurements, however, makes it difficult to constrain the problem well. This is compounded by issues with how snowpack chemistry is used (incorrectly, I believe) to quantify the amount of total nitrate in the atmosphere and the impacts of coarse particle reactive uptake of HNO3 on submicron NH4NO3 formation.

My specific comments follow:

1. p.2, line 27: I think it is fairer to say that automotive and industrial processes have increasingly been recognized as important ammonia sources in urban areas.

2. p. 5, line 31: the authors need to justify their choice of a density of 1.0 g/cm3 for coarse particles in calculating mass concentrations from APS measurements. This is quite low for typical coarse particle types, including the sea salt and dust considered in this manuscript.

3. p. 8, lines 16-18: the authors should provide information about the types of road de-icers used in the area surrounding their measurement site and whether they change as a function of forecast temperature. Many liquid de-icers commonly used in parts of the Rocky Mountain west include Ca or Mg. Are these used in Salt Lake City, on the UU campus? Do they change between conventional salt and liquid de-icers depending on temperature?

4. The analysis of reactive uptake lifetimes for HNO3 on p. 10 is quite interesting. The authors should extend this analysis to consider the relative HNO3 sink rates for NH4NO3 formation vs. reactive coarse particle uptake.

5. p. 11, lines 8-17: The authors here focus discussion on prior work concerning coarse particle uptake of HNO3 in coastal environments. This is interesting and relevant, but they should also cite observations of uptake in more continental environments which might be better models for the SLV. Lee et al. (doi:10.1016/j.atmosenv.2007.05.016), for example, examine the importance of coarse particle nitrate at both interior and coastal U.S. environments.

6. I have serious reservations about how the authors use snowpack chemistry observations to constrain atmospheric levels of total nitrate. As discussed in the manuscript, falling snow composition reflects some combination of in-cloud and below-cloud scavenging processes. The composition of snowpack on the ground is further affected, over time, by accumulated dry deposition. If one hypothesizes that significant nitrate is present in reacted coarse dust or salt particles, this coarse particle nitrate can be effectively scavenged by falling snow (coarse particle scavenging efficiencies are much high than scavenging efficiencies for accumulation mode particles) and is also very effectively deposited to snowpack on the ground by dry deposition via sedimentation of these large particles. $NH_4+$ particles, on the other hand, are submicron and thus have both lower scavenging efficiencies by falling snow crystals and much lower dry deposition rates to the surface than coarse nitrate particles. What this means is that a comparison of snowpack $NO_3-/NH_4+$ ratios is not at all representative of atmospheric concentration ratios of total nitrate/ammonium. The snowpack ratio, in the presence of coarse mode nitrate, will be significantly higher than is found in the ambient atmosphere. For this reason, the author's use of this snowpack ratio as a surrogate for what was in the atmosphere is incorrect and certainly biased high. Using this ratio to estimate the impact that eliminating coarse nitrate would have on PM2.5 NH4NO3 formation will, therefore, lead to a significant overestimate. I honestly don't see how the authors can get around this limitation on the utility of the snowpack composition data. I think the fact that the $NO_3-/NH_4+$ ratio is elevated in the snowpack does help the authors make the case that coarse particle nitrate is present, but I do not see how they can properly extend the comparison to argue what the total nitrate/ammonia ratio is in the ambient atmosphere.

7. Fig. 1: Suggest changing NH3(g) and associated flux arrows to another color. The orange color used could be misinterpreted by the reader as related to daytime pathways per the description in the caption.

8. Panels (a) and (b) of Figure 2. I personally found it somewhat unhelpful to see the

PM concentrations presented in mass units while the gas concentrations are given as mixing ratios. This makes it hard, for example, to compare relative amounts of gaseous and particulate NHx in panel (a).

---

## Author Comment (AC1) · 21 Aug 2020

We thank the reviewer for their constructive comments (shown in red below) and our responses and proposed changes to the text are shown in black

Anonymous Referee #1

This paper analyzes data from pollution events in Salt Lake City during cold periods when strong inversions lead to generation of PM2.5 ammonium nitrate. The analysis is limited in that there was not overlap in all instruments during periods of interest. For example, the APS measuring coarse mode size distributions was only operational during a short period of the overall study. It did coincide with a PM2.5 event, but during that event there was no PM2.5 aerosol data. This makes much of the analysis more speculative. Overall, the analysis is somewhat obtuse, especially the thermodynamic analysis, which could use more explanation and additional details. It would benefit from a more rigorous approach in which particle pH and partitioning of ammonium nitrate is discussed. There are some other clarifications needed, which are discussed below.

Is the measurement site representative of the greater SLC area? Does its elevation influence this?

The measurement site is located on the edge of the valley but remains within the cold air pool during pollution events, which can be confirmed by the TEOM and AIM-IC data. Due to the meteorological stability of the PCAP and the hourly sampling time of the AIM-IC, the chemical composition of PM2.5 measured there is considered generally representative of the PM2.5 that is present throughout the valley. Previous studies measuring at the base of the valley show similar average chemical composition (Kuprov et a. 2014).

Pg 2, Lines 15 to 16 regarding the health effects of NH4NO3. The logic is NH4NO3 drives PM2.5 mass, PM2.5 mass is associated with adverse health and is regulated, so NH4NO3 has adverse health implications and should be reduced to meet PM2.5 regulations. This is standard logic and so a reasonable statement, but if the authors really want to be precise, there is debate if NH4NO3 is toxic. The authors could easily add more details since health studies have been performed for the regions they are measuring, such as [Watterson et al., 2007]. The results are not as clear as the statement in this paper.

It is true there is an important discussion around whether NH4NO3 itself is toxic. For example, Park et al. 2018 evaluate toxicity scores for several types of particles and final minimal biological response to pure ammonium nitrate. However, the current PM2.5 regulations are still mass based; therefore, from a policy standpoint reducing NH4NO3 will reduce overall PM2.5 mass down to the NAAQS. We framed our introduction in that context.

Ref: Park, M., Joo, H.S., Lee, K. *et al.* Differential toxicities of fine particulate matters from various sources. *Sci Rep* **8,** 17007 (2018). https://doi.org/10.1038/s41598-018-35398-0

Also, why not discuss an environmental effect that can be directly linked to NH4NO3, the effect on visibility and haze due to its hygroscopicity, and what about nitrogen deposition; the latter seems especially important since this paper is really about deposition?

We have now included a discussion of the effect of NH4NO3 visibility and haze events that are characteristic of PCAPs in the region.

Pg 5 line 3, could not find the Markovic ref.  Is the chemistry of the water for the wet denuder and aerosol collector altered to help adsorb species?  This is most likely an issue for the wet denuder that is collecting both anions and cations gases. That is, is an absorbing species added to liquid or is it pure water?

Markovic et al. 2012 has been added to the reference list. In brief, the parallel-plate wet denuder had a constant flow of dilute (5 mM) peroxide solution through the membranes, which increases the collection efficiency of $SO_2$ without compromising the collection of acidic and basic gases. The aerosol collector is a steam chamber (fed by distilled and deionized water) that hygroscopically grows PM that is forced into solution when passed through a cyclone.

Ref: Markovic, M.Z., VandenBoer, T.C., Murphy J.G., "Characterization and optimization of an online system for the simultaneous measurement of atmospheric water-soluble constituents in the gas and particles phases" *J Environ Monitor*, 14 (7), 1872 – 1884, 2012.

What is the RH of the aerosol sampled by the TEOM and APS? Are the particles dry? Fig 1 is a nice schematic. It might be more insightful to plot the gases in units of ug/m3 to allow direct comparison with aerosol concentrations, instead of mixing ratios.  Although mixing ratios are more traditional. It should be noted in Fig 3 caption or the text that the total fine mode surface area spans sizes 0.54 to 2.5 um since it was measured by the APS. Is that really total fine particle surface area? Are the particles dry?

Fig 1. and Fig 3. were updated according to the reviewer's suggestion. The APS measured particles at ambient RH. The TEOM used during this study did have a heated inlet tube which aims to keep the sample stream at 50 degrees C. This is standard operating procedure for TEOMs that are EPA federally equivalent methods. We acknowledge the heated inlet likely volatilizes some of the particles resulting in a slight under-estimation of PM during PCAPs. Side-by-side comparison tests that were conducted by the Air Quality Lab at the U of U, showed that the differences between the two instruments are typically small (less than 2 ug/m3 during very strong PCAPs that exceed NAAQS).

Pg 9, the equation for uptake of HNO3 by coarse mode particles seems incorrect. It looks to be the formula for free molecular regime uptake, which does not apply to coarse mode particles. A correction factor due to diffusional resistance, ie something that can be derived from the size distribution and some transition regime mass transfer formula, like Fuchs Sutugin, see Seinfeld and Pandis, may be needed.

The reviewer is correct that the pseudo-first order uptake equation does not account for any diffusion limitations that occur with larger particles and large uptake coefficients. This is the upper limit to the potential HNO3 uptake loss rate to the coarse mode, as we specified in the manuscript.

Pg 13, line 8-9, This result is expected since . . . .  Please clarify.

Further explanation was added in the section highlighted by the reviewer. "To examine the sensitivity of the NH4NO3 system to the presence of non-volatile cations, addition of pNa+, pCa2+, pMg2+ and pK+ in ISORROPIA model runs underestimated the NH4NO3 concentrations observed. This large underestimation of NH4NO3 is driven by the implicit model assumptions that all particles measured by the AIM-IC have the same chemical composition and that non-volatile cations are associated with particle nitrate, which is unlikely to be the case. However, the model does show that inclusion of non-volatile cations has a strong influence in retaining NO3 in the particle phase."

It might be better to compare the predicted gas and particle partitioning, not NH4NO3 predicted, which is the form that is being assumed that the model is predicting (ie, the model does not output the NH4NO3 concentration it outputs NH4+ and NO3-, along with other species, so I assume the authors are simply taking the output NO3- adding the NH4+and comparing to a similar addition of data in the form of NH4+ and NO3- ).

The reviewer's comment is correct in explaining the modeled NH4NO3 output is sum of the concentrations of NH4 and NO3 predicted to be in the particle phase. Due to technical difficulties with the Anion IC (quantifies NO3 in gas and particle phase) during the first PCAP, there was no pNO3- to effectively compare making the HNO3 data for the same period also in question. Therefore, the authors are being conservative in focusing on NH4NO3 by estimating pNO3- based on measured pNH4+.

Aerosol pH seems to largely ignored in this work; pH values are never give despite it controlling the concentration of NO3-. It would be curious, for example, to know what the difference in pH is for with and without cation inclusion. This, and a focus on NO3- predicted vs observed would help interpret the results and provide more detail than the statement in the paper (line 8-9). It seems that what is being implied here is the model is simply doing an ion balance with the input anion and cations; ie if you add more cations than there will be less predicted NH4NO3 since there is some Ca2(NO3)2 (as an example). But isn't it much more complex since adding cations changes the pH which affects the partitioning of NH4+ and NO3-, (and other semivolatile inorganic species, such as Cl-….). At the higher concentrations, the simulated and measured agree. Why is this happening? Is it consistent with the interpretation of what is happening at lower concentrations? The authors might also consider the assumption of mixing state, they implicitly assuming that everything is internally mixed.

The reviewer is correct that the modelling work in the manuscript assumes a system in which the aerosol particles are internally mixed. It is true that any effective change in calculated pH would affect the partitioning of semi-volatile inorganic species, such as NH4 salts. However, nonvolatile species are not as pH sensitive. The authors have added more discussion of pNO3- being part of both semi-volatile and non-volatile salts, which is shown through the sensitivity analysis of adding non-volatile cations to the simplified modeled system. It is true for semi-volatile forms of nitrate salts, such as NH4NO3, pH can be a very useful indicator of thermodynamic partitioning. However, the non-volatile property of other nitrate salts, such as calcium nitrate, does not change with pH. Therefore, the uptake of HNO3 onto Ca/Mg/K containing particles is irreversible despite any changes in chemical composition and RH/T.

At higher concentrations, the system also increases in NH4+ heavily outweighing the cumulative concentrations of K+, Mg2+ and Ca2+, which may explain why the simulated and observed concentrations of NH4NO3 become closer in agreement.

The last part of the main text is also not clear (Pg 13, lines 11 to end of section). As I understand it, the idea here is that if all the NO3- and NH4+ found in snow (which got there by reacting with large particles that then fall out) was instead in the gas phase or in fine particles, and PM2.5 nonvolatile cations were present, (but no coarse mode), then the predicted NH4NO3 would be much higher (2x). But it is not clear what this proves. The situation is much more complex, and really requires a full model. Eg, would not the deposition by gases and fine mode particles change between these two cases. This is not considered. The deposition of the gas species of NH3 and HNO3 is x10 higher than a fine particle, in the case of no coarse mode, the gas concentrations could be much higher and so gas deposition much higher, which could limit PM2.5 NH4+ and NO3- formation. This is not considered in the analysis presented. The general conclusion that the coarse mode is a sink for Total NO3 is clear, but how that impacts PM2.5 NH4NO3 concentrations (it may be better to just talk about NO3-concentrations), is not clear from these simplistic analyses.

The purpose of this analysis was to use the composition of the snowpack to assess whether the atmospheric system contains a significant amount of nitrate that is not accounted for when considering only gas phase $HNO_3$ and $PM_{2.5}$ nitrate. Because we measure gas phase NH3 and $PM_{2.5}$ ammonium, and we can assume that there is very little coarse mode ammonium in the atmosphere, we can use the ratio of snowpack nitrate: snowpack ammonium to estimate the relative amount of nitrate that is present in the coarse mode. The composition of the snowpack (especially after the snow event) provides some indirect evidence of the sum of the gas and particle phase constituents lost to deposition.
We then perform a sensitivity study using ISORROPIA in which we ask whether this estimate of total atmospheric nitrate would lead to higher amounts of secondary ammonium nitrate given the measured amount of atmospheric NHx present, and assuming that the non-volatile cations were not present. We agree that the NH4NO3 system is complex and have added more explicit description of testing how sensitive the system is to the presence of coarse particles.

The reviewer is correct in that the dry deposition rates of gases and coarse particles are higher than fine particles, and this is not accounted for in the ISORROPIA model runs. A proper accounting for the impact of reactive non-volatile cations in the coarse mode on the presence of fine mode NH4NO3 would require a full 3D chemical transport model with interactive sources and sinks. However, in the absence of this type of simulation, we believe our sensitivity analysis demonstrates that the uptake of nitrate by coarse mode particles has the ability to reduce fine mode particle loading.

---

## Author Comment (AC2) · 22 Aug 2020

Hrdina et al. present an interesting, although circumstantial, case for the importance of coarse particles as a sink of HNO3 in the wintertime in the Salt Lake Valley (SLV). The authors do not have direct measurements of coarse particle composition, but make use of PM2.5 composition measurements, coarse particle size distributions, and snowpack chemistry to make a case for reactive uptake of HNO3 on coarse particles influencing the budget of HNO3 in the SLV and associated implications for fine particle chemistry. The manuscript is overall well written and most of the analyses presented sound. Some of the authors' planned measurements have missing data (e.g., the PM2.5 an-ions during the main PM episode studied). This, along with the absence of any coarse particle PM composition measurements, however, makes it difficult to constrain the problem well. This is compounded by issues with how snowpack chemistry is used (incorrectly, I believe) to quantify the amount of total nitrate in the atmosphere and the impacts of coarse particle reactive uptake of HNO3 on submicron NH4NO3 formation.

My specific comments follow:

1. p.2, line 27: I think it is fairer to say that automotive and industrial processes have increasingly been recognized as important ammonia sources in urban areas.

The role of these sources in urban areas has been added to the manuscript.

2. p. 5, line 31: the authors need to justify their choice of a density of 1.0 g/cm3 for coarse particles in calculating mass concentrations from APS measurements. This is quite low for typical coarse particle types, including the sea salt and dust considered in this manuscript.

The reviewer is correct. Typically, you would assume a density of approximately 2.0 g cm$^{-3}$. For example, Peters (2006) applied simple assumptions of shape factor and density (shape factor =1.4, density =2.0 g cm$^{-3}$) to estimate the mass concentration of ambient coarse mode particulate (PM10-2.5) with data from the Model 3321 APS. These estimates compared well with collocated, time- integrated filter based federal reference method (FRM) samplers in Phoenix (AZ) and Riverside (CA).

For this study, a constant density of 1.0 g cm$^{-3}$ was assumed in the Stokes number calculation, as a standard commonly used. The universal APS response function is applicable in most cases. However, if the particle density is greater than 2.0 g cm$^{-3}$, Wang and John (1987) found particle density affected the APS measurement. We believe thus, this assumption is valid for a surface area calculation.

In the later part of the analysis in which we make a mass balance estimate of the total amount of calcium nitrate that could be present in the coarse mode, we have modified the calculation to take into account the actual density of calcium nitrate (2.5 g cm$^{-3}$). As this was only an illustrative calculation, it does not change the interpretation presented in the manuscript.

References:

Peters T.M. (2006). "Use of the Aerodynamic Particle Sizer to measure ambient PM10-2.5: The coarse fraction of PM10." Journal of Air and Waste Management Association 56:411-416.

Wang, H.-C., and John, W. (1987). Particle Density Correction for Aerodynamic Particle Sizer, Aerosol Sci. Technol. 6:191–198.

3. p. 8, lines 16-18: the authors should provide information about the types of road de-icers used in the area surrounding their measurement site and whether they change as a function of forecast temperature. Many liquid de-icers commonly used in parts of the Rocky Mountain west include Ca or Mg. Are these used in Salt Lake City, on the UU campus? Do they change between conventional salt and liquid de-icers depending on temperature?

According to the state department, brine solution is often used before snow events as a preventative measure and solid road de-icers are generally used after, however, we could not obtain specific information about the de-icers used by the city and the UU campus. Local media reports suggest that over the last few years, ammonium nitrate deicer has been replaced with sodium acetate and sodium formate salts. In the absence of official information, we are not adding any details to the manuscript.

4. The analysis of reactive uptake lifetimes for HNO3 on p. 10 is quite interesting. The authors should extend this analysis to consider the relative HNO3 sink rates for NH4NO3 formation vs. reactive coarse particle uptake.

The rate at which HNO3 collides with fine mode particles (predominantly composed of NH4NO3) is certainly faster than with the coarse mode particles, but the HNO3-NH3-NH4NO3 system is assumed to be in equilibrium, so the net uptake, or loss, of HNO3 to NH4NO3 is zero. In the case of coarse particles, the loss of HNO3 to reactive salts is permanent. Therefore it makes more sense to limit this loss rate analysis to the coarse mode particles.

5. p. 11, lines 8-17: The authors here focus discussion on prior work concerning coarse particle uptake of HNO3 in coastal environments. This is interesting and relevant, but they should also cite observations of uptake in more continental environments which might be better models for the SLV. Lee et al. (doi:10.1016/j.atmosenv.2007.05.016), for example, examine the importance of coarse particle nitrate at both interior and coastal U.S. environments.

We have incorporated the suggested reference into the text commenting on the few studies that have speciated coarse nitrate. Lee et al. identify that coarse mode nitrate particles, formed from acid displacement, were more important in national parks areas in Arizona and Tennessee. Measurements at both sites were during Spring and Summer, respectively, so did not have competing NH4NO3, but they do highlight the fact coarse particle nitrate extend into the PM2.5 size regime and not all nitrate in this regime is associated with NH4.

6. I have serious reservations about how the authors use snowpack chemistry observations to constrain atmospheric levels of total nitrate. As discussed in the manuscript, falling snow composition reflects some combination of in-cloud and below-cloud scavenging processes. The composition of snowpack on the ground is further affected, over time, by accumulated

dry deposition. If one hypothesizes that significant nitrate is present in reacted coarse dust or salt particles, this coarse particle nitrate can be effectively scavenged by falling snow (coarse particle scavenging efficiencies are much high than scavenging efficiencies for accumulation mode particles) and is also very effectively deposited to snowpack on the ground by dry deposition via sedimentation of these large particles.  NH4+ particles, on the other hand, are submicron and thus have both lower scavenging efficiencies by falling snow crystals and much lower dry deposition rates to the surface than coarse nitrate particles. What this means is that a comparison of snowpack NO3-/NH4+ ratios is not at all representative of atmospheric concentration ratios of total nitrate/ammonium.  The snowpack ratio, in the presence of coarse mode nitrate, will be significantly higher than is found in the ambient atmosphere. For this reason, the author's use of this snowpack ratio as a surrogate for what was in the atmosphere is incorrect and certainly biased high. Using this ratio to estimate the impact that eliminating coarse nitrate would have on PM2.5 NH4NO3 formation will, therefore, lead to a significant overestimate.  I honestly don't see how the authors can get around this limitation on the utility of the snowpack composition data.  I think the fact that the NO3-/NH4+ ratio is elevated in the snowpack does help the authors make the case that coarse particle nitrate is present, but I do not see how they can properly extend the comparison to argue what the total nitrate/ammonia ratio is in the ambient atmosphere.

The reviewer raises some of the same concerns identified by referee #1 regarding the interpretation of the snowpack data. The reviewer is correct that the composition of the snowpack does not precisely and quantitatively reflect the relative abundance of the components in the atmosphere. It is true that the dry deposition and scavenging of coarse particles is faster than for fine particles, but not necessarily much faster than for the gas phase constituents. Also, during a snowfall event, the *efficiency* of scavenging may not be important if the scavenging in nearly complete.

In the absence of the ability to directly quantify coarse mode nitrate in the atmosphere, we use an approach that allows us to infer how much nitrate may be present as Ca or Na salts in the coarse mode. We agree with the reviewer that this approach likely reflects an upper estimate of its importance to the atmospheric burden (though not to the overall nitrate budget, since the deposition rate matters). Our subsequent simple analysis then examines how much additional NH4NO3 could have been formed if no reactive coarse particles had been present. We have amended the text to clarify this sensitivity test and to explain how this reflects an upper estimate, as pointed out by the reviewer.

7. Fig. 1: Suggest changing NH3(g) and associated flux arrows to another color. The orange color used could be misinterpreted by the reader as related to daytime pathways per the description in the caption.8.  Panels (a) and (b) of Figure 2.  I personally found it somewhat unhelpful to see the PM concentrations presented in mass units while the gas concentrations are given as mixing ratios. This makes it hard, for example, to compare relative amounts of gaseous and particulate NHx in panel (a).

We have updated the figures as suggested.

---

## Referee Report (RR1)

Thank you to the author for being responsive to many of the comments provided on the original manuscript. I have only a couple comments that should be addressed in finalizing the manuscript.

1. The authors provided the response shown below to a comment in the initial review, however, the changes they claim to have made - to add discussion about coarse nitrate in non-coastal environments - were not actually included in the tracked comments version of the revised manuscript. The text in this section on p. 11 of the revised manuscript remains the same as that in the original manuscript. No change was actually made. I assume this is just an oversight in preparing the revised manuscript.

Original review comment 5. p. 11, lines 8-17: The authors here focus discussion on prior work concerning coarse particle uptake of HNO3 in coastal environments. This is interesting and relevant, but they should also cite observations of uptake in more continental environments which might be better models for the SLV. Lee et al. (doi:10.1016/j.atmosenv.2007.05.016), for example, examine the importance of coarse particle nitrate at both interior and coastal U.S. environments.

Author response. We have incorporated the suggested reference into the text commenting on the few studies that have speciated coarse nitrate. Lee et al. identify that coarse mode nitrate particles, formed from acid displacement, were more important in national parks areas in Arizona and Tennessee. Measurements at both sites were during Spring and Summer, respectively, so did not have competing NH4NO3, but they do highlight the fact coarse particle nitrate extend into the PM2.5 size regime and not all nitrate in this regime is associated with NH4.

2. I appreciate that the authors have now somewhat caveated the use of the snowpack nitrate:ammonium ratio as a proxy in estimating how much nitrate in the atmosphere was present in coarse particles. I do not see convincing evidence, however, to support their claim that "this overestimate is within reason." Nor is a reader likely to understand what is meant by a "within reason" error estimate.

In the original review I point out that the snowpack composition nitrate:ammonium ratio will be biased high by the greater scavenging efficiency during precipitation and the higher dry deposition velocity to snow after precipitation of coarse nitrate particles vs. fine ammonium particles. The authors' responses that
   (a) scavenging efficiency differences are not important if scavenging is complete ignores the fact that scavenging is generally not complete for fine particles and we, in any case, have no information about the extent of scavenging in the snowfall episodes studied
   (b) coarse particle dry deposition may not be much faster than gas species dry deposition adds even further uncertainty about the relationship between snowpack composition and airborne nitrate:ammonium ratios since one has coarse and fine particles and gaseous nitric acid and ammonia all dry depositing to the snowpack at differing rates that depend on widely different deposition velocities and concentrations.

I ask that the authors provide the reader with a more complete discussion of these significant limitations to their use of snowpack composition when they provide a quantitative estimate of the potential increase of NH4NO3 in the absence of coarse particles.